# Validation of wind measurements of two MST radars in northern Sweden and in Antarctica

Evgenia Belova[1], Peter Voelger[1], Sheila Kirkwood[1], Susanna Hagelin[2], Magnus Lindskog[2], Heiner Körnich[2], Sourav Chatterjee[3], and Karathazhiyath Satheesan[4]

[1]Swedish Institute of Space Physics, Kiruna, SE-98128, Sweden
[2]Swedish Meteorological and Hydrological Institute, Norrköping, SE-60176, Sweden
[3]National Centre for Polar and Ocean Research, Ministry of Earth Sciences, Goa, 403804, India
[4]Department of Atmospheric Sciences, School of Marine Sciences Cochin University of Science and Technology, Cochin, Kerala, 682 016, India

*Correspondence to*: Evgenia Belova (evgenia.belova@irf.se)

**Abstract.** Two atmospheric VHF radars: ESRAD located near Kiruna in the Swedish Arctic and MARA at the Indian research station Maitri in Antarctica perform wind measurements in the troposphere and lower stratosphere on a regular basis. We compared horizontal winds at altitudes between about 0.5 km and 14 km derived from the radar data using the full correlation analysis (FCA) technique with radiosonde observations and models. The comparison with 28 radiosondes launched from January 2017 to August 2019 showed that ESRAD underestimates the zonal and meridional winds by about 8% and 25 %, respectively. This is likely caused by the receiver group arrangement used for the FCA together with a high level of non-white noise. A similar result was found when comparing with the regional NWP model HARMONIE-AROME for the period September 2018 – May 2019. The MARA winds were compared with winds from radiosondes for the period February - October 2014 (291 occasions). In contrast to ESRAD, there is no indication that MARA underestimates the winds compared to the sondes. The mean difference between the radar and radiosonde winds is close to zero for both zonal and meridional components. The comparison of MARA with the ECMWF ERA5 reanalysis for January – December 2019 reveals good agreement with the mean difference between 0.1 m/s and -0.5 m/s depending on the component and season. The random errors in the wind components (standard deviation over all estimates in 1-hour averages) are typically 2-3 m/s for both radars. Standard deviation of the differences between radars and sondes are 3 - 5 m/s.

## 1 Introduction

Atmospheric winds are an essential part of weather and climate, however atmospheric measurements are skewed towards temperature, moisture or pressure (WMO, 2012). This skewness results from the fact that winds are more difficult to measure remotely. Atmospheric radars have been used for wind measurements since the 1950s. The history, design, methods and applications of atmospheric radars are described in the comprehensive book by Hocking et al. (2016). The mesosphere-stratosphere-troposphere (MST) radar ESRAD located near Kiruna in the Swedish Arctic has been in operation since 1996 (Chilson et al., 1999). It has run continuously (with the exception of a few short breaks due to technical problems) and

delivers three components of wind. Another wind profiler MARA has been operated at various locations in Antarctica since 2006 (Kirkwood et al., 2007). In some years MARA was able to run for only a few months (due to stations being closed or experiencing severe weather conditions), in other years 12 months of operations have been possible. In August 2018 ESA launched the Earth explorer satellite Aeolus with the main objective to provide wind profiles in the troposphere and lower stratosphere (0-30 km altitudes) with global coverage (ESA, 2018; Straume et al., 2019). The satellite mission was specifically designed to address the lack of wind profile observations in many parts of the globe, such as the Tropics and over the oceans. Both radars, ESRAD and MARA, are involved in the Aeolus calibration and validation activities, and scarcity of data at high latitudes makes these radar observations very valuable for validation of Aeolus wind products in these regions. Before making a validation of Aeolus winds we need to evaluate carefully the accuracy of the wind measurements made with the radars themselves. This can be done in comparison with other measurements and with established models. In this paper we aim to validate the ESRAD and MARA winds in the troposphere and lower stratosphere by comparison with winds observed with radiosondes, with the regional HARMONIE-AROME model and with the ECMWF ERA5 reanalysis for the period following the Aeolus launch.

## 2 ESRAD

### 2.1 Wind measurements

The Esrange MST radar (ESRAD) is an atmospheric radar located at Esrange (68°N, 21°E) in northern Sweden. It is a joint venture between the Swedish Institute of Space Physics (IRF) and Swedish Space Corporation (SSC) Esrange Space Center. ESRAD began operations in July 1996 and had two major upgrades in 2004 and 2015. The purpose of the radar is to provide information on the dynamic state of the atmosphere – winds, waves, turbulence and layering, from the troposphere up to the mesopause (ca. 0.5-90 km altitude). It operates at 52 MHz and the nominal peak transmit power is 72 kW, however only 30 kW is available at present due to progressive failure of several power blocks. The ESRAD main antenna array, consisting of 288 five-element Yagis, is divided into 12 identical groups each connected to one power block and to a separate receiver. The receivers have 1 MHz bandwidth and separate detection of in-phase and quadrature components. This allows post-detection beam-steering and full spectral analysis of the return signal. The radar transmits vertically with the whole main antenna array, but for reception one can use 12 segments in different combinations. In 2015 a small separate receive-only array (3 sub-arrays of 4 Yagis, three-element each) was constructed about 30 m away from the south-east corner of the main array. In combination with transmitting on only part of the main array, this allows measurements at the lowest altitudes starting at about 0.5 km. However due to intermittent time synchronization errors, we do not use the data from this array in the present paper. The parameters of ESRAD are presented in Table 1 and a diagram of the antenna array is shown in Fig. 1. Vertical wind is derived from the Doppler shift of the return signal by combining (coherently) the data from all receivers in groups 1-12. The concept behind the radar horizontal wind measurements is the following. A radar transmits electromagnetic waves that are scattered or reflected from inhomogeneities in the atmospheric refractive index. An ensemble of such

inhomogeneities in an atmospheric layer works as a diffraction filter that creates a diffraction pattern of return signal on the
ground which can be measured by spaced receivers (antenna segments). Scatterers of the radar wave are advected by wind, and it has been shown that the diffraction pattern moves along the ground with double the wind velocity (Briggs, 1980).

| Radar | ESRAD | MARA |
|---|---|---|
| Geographical coordinates | 68°N 21°E | 71°S 12°E |
| Height above sea level | 295 m | 117 m |
| Frequency | 52 MHz | 54.5 MHz |
| Peak power | 72 kW nominal (30 kW now) | 20 kW |
| Antenna effective area | 3740 m$^2$ | 540 m$^2$ |

**Table 1: Characteristics of ESRAD and MARA radars.**

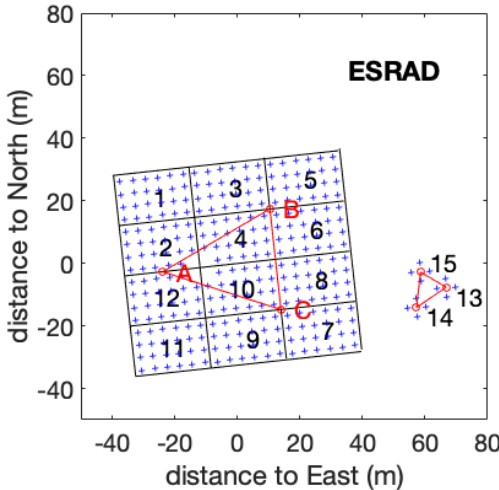

**Figure 1: Configuration of the ESRAD antenna field. Each blue cross marks the position of a Yagi antenna in the main array (groups 1-12) and in the 'remote' groups (13-15). Each group 1-15 is connected to a separate receiver. Groups 1-12 are also connected to transmitters.**

Horizontal winds are derived by using cross-correlation technique to find the time it takes for the diffraction pattern of the
irregularities to pass the different antenna sub-arrays, corrected for the irregularity decay time. This method is known as full correlation analysis (FCA) and was developed by Briggs et al. (1950) and Briggs (1984). For ESRAD we adopted the FCA algorithm as described by Holdsworth (1995).   The FCA is one of two commonly used radar techniques for atmospheric horizontal wind estimation (Hocking et al., 2016). The other is the Doppler Beam Swinging (DBS) method, which is not technically applicable for our radar.

| Experiment name | fca_150 | fca_900 | fcx_aeolus |
|---|---|---|---|
| Pulse Repetition Frequency, Hz | 4688 / 3125** | 1300 | 2490 |
| Code | none | none | none |
| Number of coherent integrations* | 512 / 896 | 256 | 512 |
| Duration of measurements, s | 120 | 120 | 120 |
| Pulse length, μs | 1 | 6 | 6 |
| Pulse shape | shaped trapezoid | shaped trapezoid | shaped trapezoid |
| Receiver filter, MHz | 1 | 0.250 | 0.250 |
| Start height, m | 150 | 1050 | 1050 |
| Stop height, m | 29100 | 100650 | 27450 |
| Number of height gates | 194 | 167 | 45 |
| Height sampling /resolution, m | 150 / 150 | 600 / 900 | 600 / 900 |

Table 2: Parameters of the ESRAD experiments used in the paper. *This is the total number of integrations, including those applied in analysis, for heights up to 16 km. ** summer/winter

Basic software for radar control and data acquisition from the radar manufacturer Genesis Sofware Pty as well as our own software for analysis run in real-time. The radar runs continuously, cycling between experiments optimized for the lower troposphere, troposphere/stratosphere, or mesosphere. A typical cycle measures for 1-2 minutes in each mode, repeating every 3-6 minutes. Special cycles, optimized for specific goals may be run from time to time, for example in this paper we use data from a special experiment fcx_aeolus designed in support of the ESA Aeolus satellite mission in addition to two common experiments fca_150 and fca_900. We run a sequence of four experiments (one of them is not used in the paper) for two minutes each thus providing wind data every eight minutes. The parameters of the experiments are listed in Table 2 and the arrangement of the receivers is shown in Fig. 1. For the full correlation analysis from the main array, digitised data from sets of 4 groups are added coherently in software to improve the signal-to-noise ratio to make 'supergroups' with centres at A (groups 1, 2, 11, 12), B (groups 2, 4, 5, 6) and C (groups 7, 8, 9, 10). The red triangle ABC indicates the corresponding baselines for the full correlation analysis. More detailed descriptions of ESRAD can be found in Chilson et al. (1999) and Kirkwood et al. (2010).

**2.2 ESRAD versus radiosondes**

We use the wind data from 28 radiosondes (ascents) that were launched from Esrange during the period of January 2017 - August 2019. The radiosondes have been launched as support for different balloon and rocket campaigns held at Esrange. Standard GPS radiosondes from the Vaisala company were used, typically reaching 20 -30 km heights. The raw data were sampled at 2 s intervals, resulting in an uneven vertical interval, which varies from 6 to 9 m.

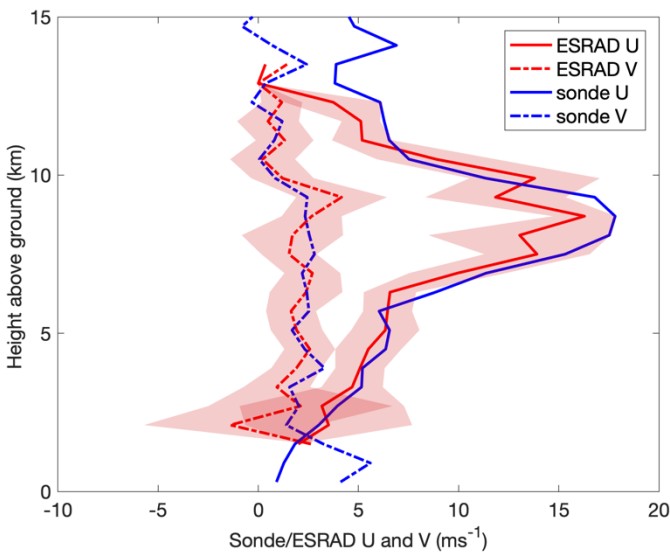

**Figure 2: Vertical profiles of the zonal U and meridional V components of wind measured by the ESRAD radar and radiosonde on 15 August 2018. Shading indicates one standard deviation of the ESRAD winds.**

An example of the zonal and meridional wind profiles as measured by the ESRAD radar and by a radiosonde on 15 August 2018 is shown in Fig. 2. The ESRAD data for three experiments listed in Table 2 were averaged over the 1-hour interval centred on the radiosonde launch time. The radar and sonde wind data were averaged to the same altitude bins starting from 300 m with 600 m resolution. We see that for the altitude range from about 1.5 km to about 13 km the radar winds are in good agreement with radiosonde ones, at least within one standard deviation (the standard deviation refers to the distribution of individual radar estimates for all of the times, heights and experiments in the averaging bins).

We did the same averaging for all 28 occasions when radiosondes were launched and the results for zonal and meridional winds are presented in Fig. 3. Our comparisons are focused on U and V components because they will be further used for evaluation of the Aeolus horizontal line-of-sight winds. We also plot in Fig. 3 the linear fits as dotted-dashed lines: the radar on the sondes in blue and the sondes on the radar in green. A robust fitting with bisquare weights was used in order to reduce the contribution of outliers. Two fits were done because the radar and sondes both measure winds with different uncertainties that we do not know absolutely (e. g. additional errors can be due to temporal and spatial separations of the instruments). Then "the best fit" between data from these instruments will be somewhere between these two fits. We do not determine its exact parameters as proposed by Hocking et al. (2001) because both regression lines lie rather close to each other.

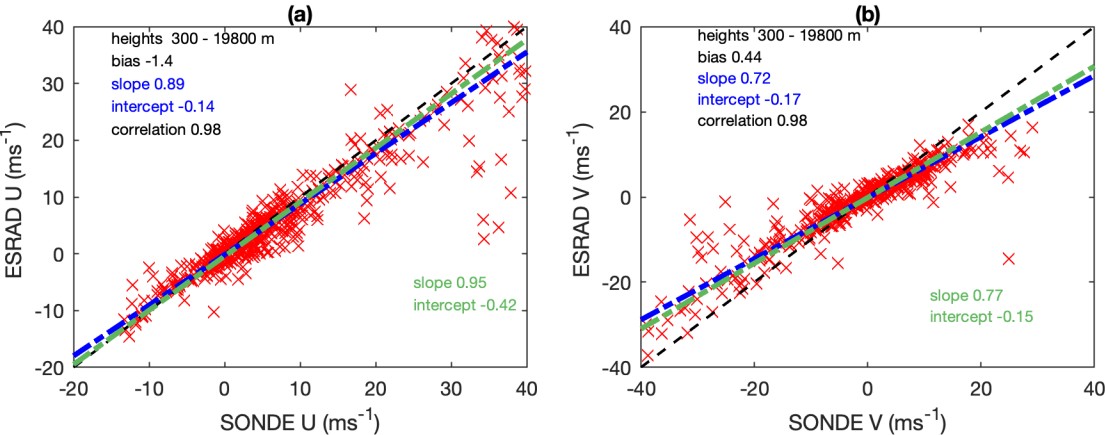

**Figure 3: Comparison of the ESRAD and radiosonde (a) zonal and (b) meridional winds. The linear fits are shown as dashed-dotted lines: the radar on sondes in blue and the sondes on the radar in green. The black dashed straight line corresponds to the case when the radar velocity is equal to the sonde velocity. More details of the legend in the inserts are in the text.**

The parameters of the linear fits such as slope and intercept are shown in Figure 3 with the same colour as the corresponding lines. The slope is significantly closer to 1 for the zonal wind fit than for meridional one, all intercepts are smaller than 0.5 m/s. We also calculated a mean difference between the radar and radiosonde winds, it is nominated as 'bias' and shown in the inserts in the figure. The mean difference for U and V wind components is -1.4 m/s and 0.4 m/s, respectively, and the slopes are less than 1, which implies that the radar underestimates wind compare to the radiosonde. The correlation

coefficient between radar and sonde data is 0.98 for both zonal and meridional wind. Behaviour of the inter-comparison parameters as a function of height is shown in Fig. 4. From this figure we see that the parameters vary irregularly with height, however the correlation coefficient and slope of fit tends to decrease with increasing heights, while absolute values of the mean difference for both wind components increase with height. The largest differences between the radar and radiosondes are observed at the lowest and highest altitudes. The former can be explained by poor radar performance at the

lower heights, the latter may be due to increased spatial separation between the radar and radiosonde sampling volumes. These higher altitudes will also show larger deviations for the same % underestimate as winds are stronger there, as seen in Fig. 3. For altitudes above about 2 km and below about 12 km, where there is a high enough number of the data for comparison, the agreement between the radar and radiosondes is good similarly as was shown for one day in Fig.2. The random errors do not vary significantly with altitude (not shown).

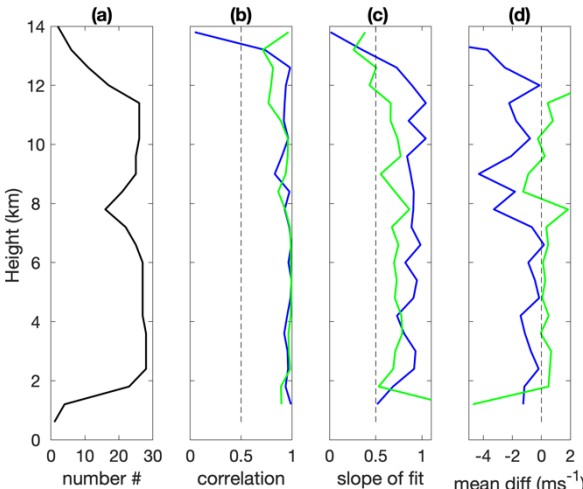

**Figure 4:** Altitude profiles of (a) the number of ESRAD and radiosonde velocities available for the comparison, (b) correlation coefficient between them, (c) slope of the radar-on-sondes linear fits and (d) mean difference between the radar and radiosonde winds. Blue and green colours indicate zonal and meridional wind, respectively.

The mean standard deviation of the radar winds (from the distributions of individual wind estimates in each averaging bin is 2.3 (2.0) m/s for zonal (meridional) components. To quantify the random error in the differences between sonde and radar winds, we first correct the ESRAD winds for the systematic underestimate in wind components (by 25% in meridional and 8% in zonal components). The standard deviation of the difference between radar (corrected) and sonde winds is 4.4 (4.8) m/s. This is a combination of uncertainties in both radar and sonde measurements, differences due to the differing locations of the measurements, and differences between instantaneous (sonde) measurements and 1-hour averaged radar measurements.

## 2.3 ESRAD versus the HARMONIE model

In order to validate the radar wind over an extended, continuous period of time we made the comparisons with winds produced using the HARMONIE-AROME km-scale NWP model (Bengtsson et al., 2017). It is one configuration of the shared Aire Limitée Adaptation Dynamique Developpement InterNational (ALADIN)-High-Resolution Limited-Area Model (HIRLAM) NWP system, developed jointly by 26 countries in Europe and northern Africa. HARMONIE-AROME is comprised of a data assimilation system for the surface and upper-air together with an atmospheric forecast model, including the SURFEX surface scheme (Masson et al., 2013). To provide the best possible initial model state for the surface and atmosphere, a data assimilation is applied. The surface data assimilation is based on optimal interpolation (Giard and Bazile, 2000) while a 3-dimensional variational data assimilation scheme is used for the upper atmosphere (Fischer, 2005). Operational ensemble forecasts are produced within the collaboration MetCoOp (Meteorological Co-operation on Operational NWP), including the national meteorological services of Sweden, Norway, Finland and Estonia (Müller et al., 2017). The operational domain covers Fenno-Scandinavia and has 960 x 1080 horizontal grid points with a resolution of 2.5

km for each of the 65 vertical levels. The model top is at approximately 10 hPa and the vertical model level separation is about 50 m close to the surface and up to 1 km in the stratosphere.

We looked at the period from 1 September 2018 to 31 May 2019. The choice was motivated by changes in operation of the Aeolus satellite - during this interval the Doppler lidar on the board of Aeolus satellite used laser A (it was switched to laser B in June 2019). Again, the ESRAD winds were averaged over three experiments, over 1-hour centered on 00 UT, 06 UT, 12 UT and 18 UT, which are the times of the model output, and over 1-km altitude gates starting from the ground. Then model winds at the grid point closest to ESRAD were interpolated for the same altitudes.

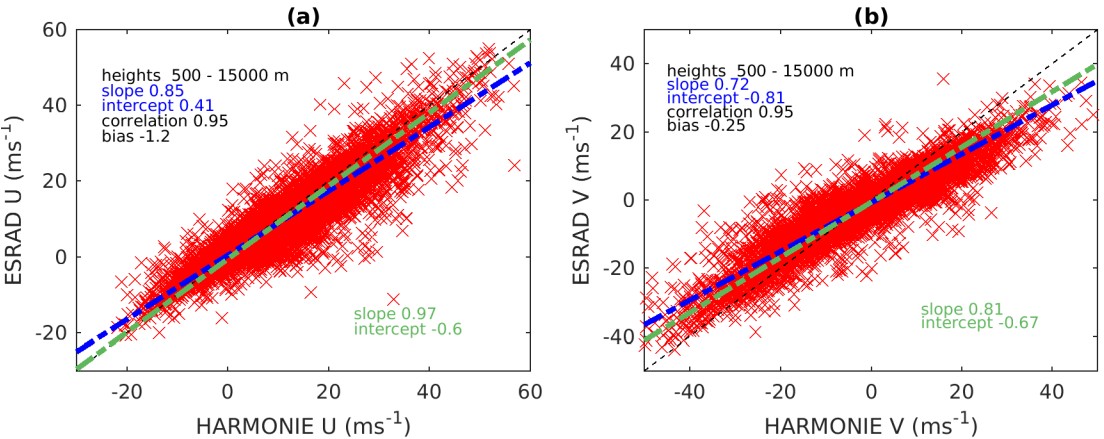

Figure 5: Comparison of the ESRAD and HARMONIE model (a) zonal and (b) meridional winds for period of September 2018 – May 2019. The designations are the same as for Fig. 3.

Before making a comparison for all nine months we looked at seasonal behaviour of winds at altitudes from 5 to 15 km at the ESRAD site using the European Centre for Medium-Range Weather Forecasts (ECMWF) reanalysis. On the basis of the horizontal wind speed and direction averaged over 2005-2016 (not shown) we can distinguish two seasons when winds show different behaviour: from September to April and from May to August. We decided to group our data altogether because only one month (May) belongs to another season. The ESRAD zonal and meridional winds versus the HARMONIE corresponding winds are shown in Fig. 5, where all data for nine months are presented and the linear fits are drawn. In general, there is a good agreement between the radar and model winds, however it is better for the zonal component than for the meridional one. As in comparison with radiosondes, ESRAD underestimates both wind components compared to the HARMONIE: the slopes for the zonal wind fits are 0.85/0.97 and mean difference is -1.2 m/s whereas they are 0.72/0.81 and -0.3 m/s, respectively, for the meridional wind. We also computed the slope of fit of the radar on the model, their correlation and mean difference as a function of height and presented in Fig. 6. At altitudes above about 2 km the agreement between the radar and the model is very good with an average correlation of 0.95. Below 2 km the ESRAD winds appear to be poorly correlated with the HARMONIE winds, similarly as in comparison with the radiosondes (Fig. 4). The radar random error

variation with height is 1.9-3.3 m/s for the meridional wind and 2.3-3.7 m/s for the zonal wind (not shown). The mean standard deviation of the radar winds (from the distributions of individual wind estimates in each averaging bin is 2.8 (2.4) m/s for zonal (meridional) components. The radar standard deviation variation with height is 1.9-3.3 m/s for the meridional wind and 2.3-3.7 m/s for the zonal wind (not shown). The standard deviation of the difference between (corrected) radar and HARMONIE winds is slightly higher, 4.3 (4.9) m/s, but very close to the values found in the comparison with radiosondes.

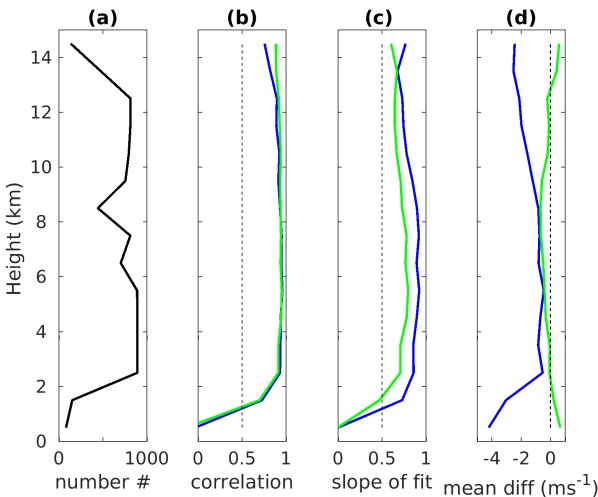

**Figure 6: Altitude profiles of (a) the number of ESRAD and HARMONIE velocities available for the comparison, (b) correlation coefficient between them, (c) slope of the radar-on-model linear fits and (d) mean difference between the radar and model winds for period of September 2018 – May 2019. Blue and green colours indicate zonal and meridional wind, respectively.**

## 3 MARA

### 3.1 Description of the radar

MARA (Moveable Atmospheric Radar for Antarctica) is a 54.5 MHz wind-profiler type radar. It is in many ways a smaller, movable clone of ESRAD (Kirkwood et al., 2007). MARA is less powerful than ESRAD, having peak power of 20 kW. The antenna consists of 3 adjacent square arrays, each with 16 tuned dipoles with reflectors (see Table 1 for the main parameters of MARA). The arrangement of the antenna array is shown in Figure 7. There the red triangle 123 indicates the baselines for the full correlation analysis for the main array. The 'remote' groups 4, 5, 6 are used for very low heights where useful data 195 cannot be obtained from transmitting groups. Common experimental modes and analysis are the same or very similar for the ESRAD and MARA radars. In Table 3 the parameters of the MARA experiments used in this study are presented. Starting in 2006, MARA has been operated at various locations in Antarctica. Since 2014 it has been located at the Indian research station Maitri (71°S, 12°E) (http://www.ncaor.gov.in/antarcticas/display/376-maitri-) and in November 2017 IRF transferred the ownership of MARA to the National Centre for Polar and Ocean Research, India. Weather conditions at Maitri so far

have been very harsh for MARA's antenna hardware which leads to interruptions in the MARA observations, with sometimes long breaks since repairs are only possible during the Antarctic summer.

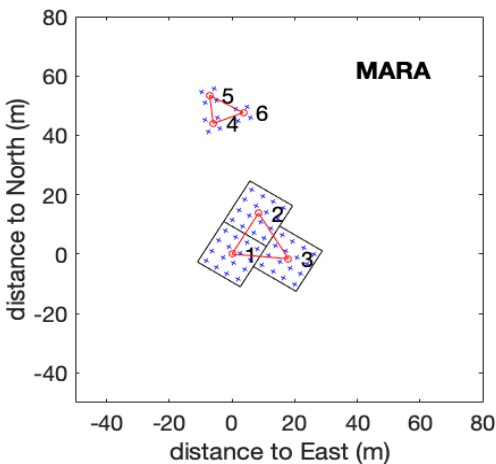

**Figure 7: Configuration of the MARA antenna field at Maitri station, Antarctica. Each blue cross marks the position of an antenna, single polarisation, dipoles with reflectors in the main array (groups 1-3) and 3-element Yagis in the 'remote' groups (4-6). Each group is connected to a separate receiver. Groups 1-3 are also connected to transmitters. The red triangles indicate the baselines for the FCA.**

| Experiment name | fca_75 | fcw_150 | fca_4500 |
|---|---|---|---|
| Pulse Repetition Frequency, Hz | 10300 | 10300 | 1300 |
| Code | none | none | 8-bit complementary |
| Number of coherent integrations* | 2048 | 2048 | 128 |
| Measurement duration, s | 60 | 60 | 60 |
| Pulse length, μs | 0.5 | 1 | 8 x 4 |
| Pulse shape | Gaussian | Gaussian | shaped trapezoid |
| Receiver filter, MHz | 1.000 | 0.500 | 0.250 |
| Start height, m | 100 | 100 | 4800 |
| Stop height, m | 6200 | 13500 | 104400 |
| Number of height gates | 123 | 135 | 167 |
| Height sampling / resolution, m | 50 / 75 | 100 /150 | 600 / 600 |

**Table 3: Parameters of the MARA experiments used in the paper. *This is the total number of integrations, including those applied in analysis, for heights up to 40 km.**

## 3.2 MARA versus radiosondes

After MARA was deployed at Maitri in 2014, the radar winds were validated using radiosondes launched from the nearby (4 km to the east) Russian station Novolazarevskaya. However, since July 2018 the radio soundings have been interrupted and have not started again so far. We present here comparison of MARA with radiosondes launched between 08 February 2014 and 30 October 2014 (291 occasions). Radiosonde winds were retrieved from the international database at Univ. Wyoming (http://weather.uwyo.edu/upperair/sounding.html). On average, radiosonde winds were available at 21 heights between the limits (700 - 11000 m) suitable for comparison with MARA. Sondes were usually launched at 0 UT each day, occasionally also at 12 UT and are compared with 1-hour wind averages 00-01 UT (or 12-13 UT) from MARA, including all estimates where the height of the sonde wind was within the height resolution of the radar wind. Full correlation analysis 'true' winds from each of the three experiments (Table 3) and both main and remote antenna groups are used, with usual acceptance criteria applied, providing on average 38 comparison points per sonde. The results are presented in Fig. 8.

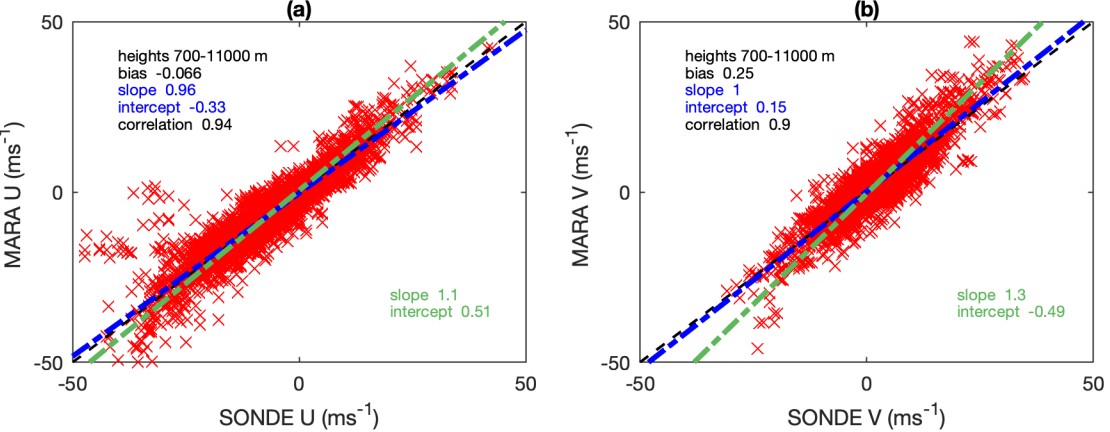

**Fig. 8: Comparison of the MARA and radiosonde (a) zonal and (b) meridional winds. The linear fits are shown as dashed-dotted lines: the radar on sondes in blue and the sondes on the radar in green. The black dashed straight line corresponds to the case when the radar velocity is equal to the sonde velocity.**

We also plot there the linear fits as in Fig. 3, and the parameters of the fits together with the bias and correlation are provided in the inserts. In contrast to ESRAD, there is no indication that MARA underestimates the winds compared to the sondes (the slopes of the fits for MARA on sonde are slightly less than 1, for sonde on MARA, slightly more than 1). The bias, defined as the mean difference between the radar and radiosonde winds, is close to zero for both zonal and meridional components. The mean standard deviation of the radar winds (from the distributions of individual wind estimates in each averaging bin is 2.1 (1.5) m/s for zonal (meridional) components. The standard deviation of the difference between radar and sonde winds is higher, 3.7 (2.9) m/s. This can be due to random errors in the sonde winds, the differing locations of the

measurements and differences between instantaneous winds (sondes) and height/time averages (radar). The parameters of the inter-comparison do not vary significantly with height (not shown).

### 3.3 MARA versus ECMWF ERA5

Because of lack of most recent radiosonde data close to Maitri we also compare the MARA winds with those from the ECMWF reanalysis ERA5 (Hersbach et al., 2020) for 2019 when the Aeolus satellite has been in orbit. The data cover the Earth on a 30-km grid and resolve the atmosphere using 137 levels unequally spread from the surface up to 1 Pa pressure
level at about 80 km altitude. We use 1-hourly data for the altitude range 0-20 km at the grid point closest to the Maitri location, from January until December 2019, when the MARA data were available.  We divided data into two groups: 1st - from March to September, 2nd – January, February, October, November and December. This corresponds to generally different behaviour of winds over Maitri as seen from the ECMWF data (not shown here). Plots of MARA versus ERA5 for the zonal and meridional winds as well as the linear fits for these two intervals are presented in Figures 9 and 10. In general,
there is good agreement between the radar and model for both intervals. The best linear fits, which lie somewhere between the green and blue lines, have likely a slope close to or less than 1. This implies the radar slightly underestimates horizontal wind compared to the model. The correlation is high (92-95%) and the biases are small (< 0.5 m/s) and negative (with one exception).  The correlation is higher and the slope is closer to 1 for the zonal component compared to the meridional one. There are no essential distinctions between the statistics for the two intervals, while the range of velocity values changes
from one period to another and the bias of the meridional wind changes the value from small positive to small negative. Additionally, there are visually more outliers for data from March to September 2019. The mean standard deviation of the radar winds (from the distributions of individual wind estimates in each averaging bin) is 2.6 (2.1) m/s for zonal (meridional) components, and they are about the same for both intervals. The standard deviation of the difference between radar and ERA5 winds is higher, 4.0 (3.2) m/s for October-February, 4.5 (4.2) m/s for March-September. This is also slightly higher
than for the comparison with sondes in Section 3.2, particularly for the meridional wind during the winter period March-September. This likely points to limitations in ERA5 at the MARA location.

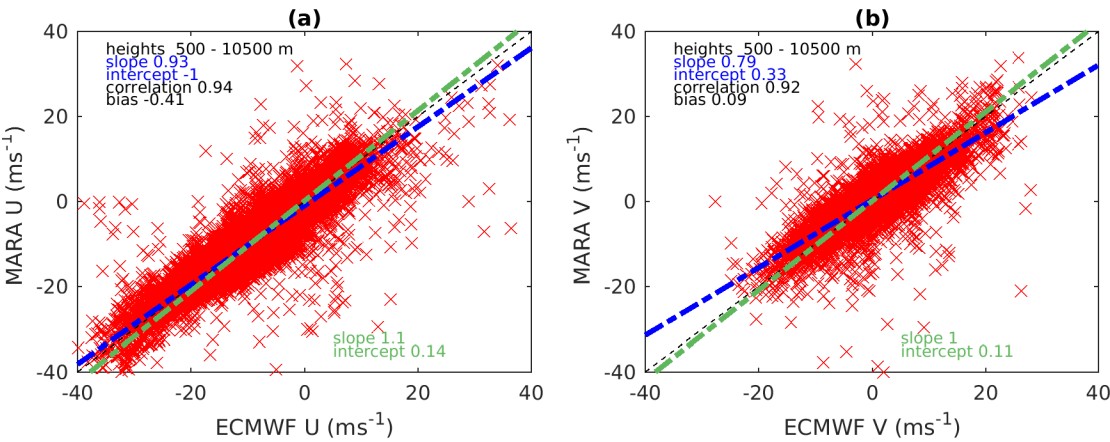

**Figure 9: Comparison of the MARA and ECMWF ERA5 model (a) zonal and (b) meridional winds for period of January, February, October – December 2019. The designations are the same as for Fig. 3.**

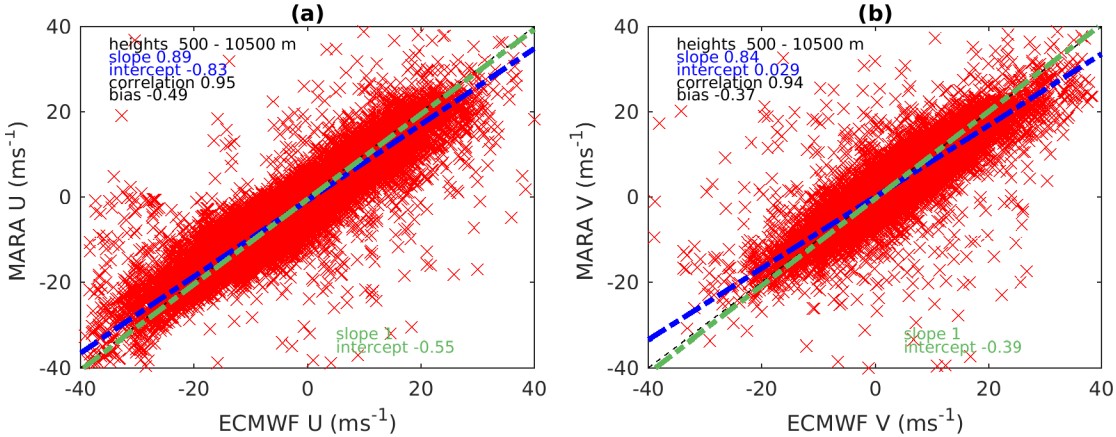

**Figure 10: The same as Figure 9 but for period of March – September 2019.**

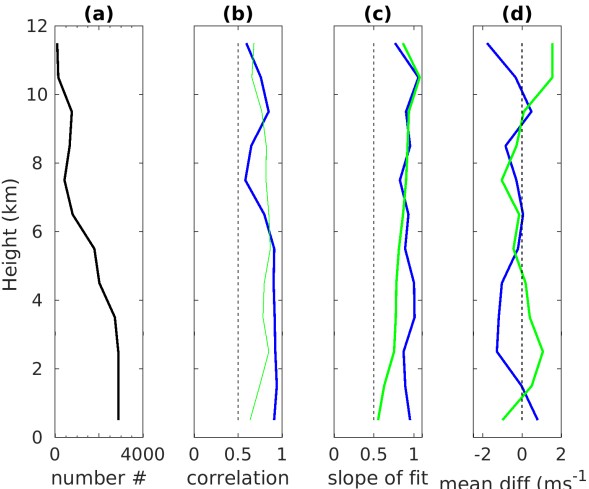

**Figure 11: Altitude profiles of (a) the number of MARA and ERA5 velocities available for the comparison, (b) correlation coefficient between them, (c) slope of the radar-on-model linear fits and (d) mean difference between the radar and model winds for January, February, October – December 2019. Blue and green colours indicate zonal and meridional wind, respectively.**

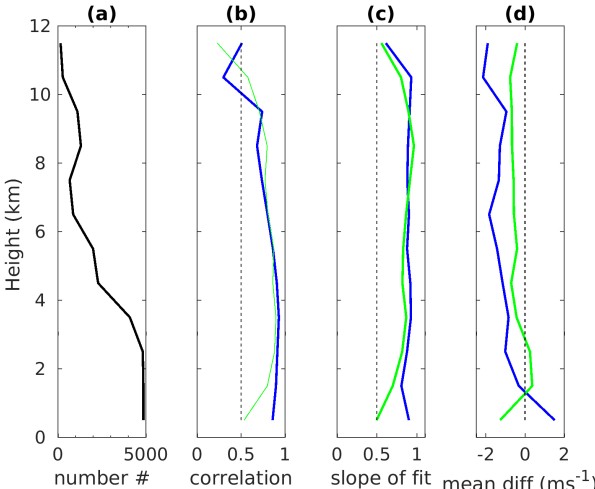

**Figure 12: The same as Fig. 11 but for March – September 2019.**

In Fig. 11 and Fig. 12 we present the vertical profiles of the inter-comparison statistics for two periods. The agreement

between the radar and model is good for all heights from 500 m until 10.5 km above which there are not so many radar data, radar-model correlation weakens and absolute values of biases increase. The difference between the two periods under consideration is only seen in the altitude profiles of the radar-model biases: they vary from negative to positive for the 1st period and are negative (with a few exceptions) for the 2nd period. The standard deviation of the MARA wind (of the

samples averaged in each height bin) varies between a maximum of 3.4 m/s at 0.5 km to a minimum of 2 m/s at 6 km
attitude (not shown). Significantly more wind data for the lower heights (< 2 km) are available for MARA than for ESRAD.

## 4 Discussion

Several studies have been published on inter-comparison of windprofiler and radiosondes, models and different radar techniques for deriving winds (e. g. Vincent et al. 1987; Gage et al. 1988; Kudeki et al. 1993; MacKinnon 2001; Stober et al. 2012). Some of them were reviewed by Reid et al. (2005) where the authors also presented their own comparison of the
Mount Gambier wind profiling radar in Australia using FCA with 3000 radiosondes. The authors confirmed the other studies and found that the FCA winds underestimate in magnitude by about 3-7 % relative to the radiosonde winds in the planetary boundary layer, troposphere and lower stratosphere. The reasons given for this bias in the FCA technique are that noise and antenna coupling tend to reduce cross-correlation values, and hence, estimated wind speeds (Holdsworth, 1999). Other possible reasons of differences between profiler winds and other techniques is spatial and temporal separation between
measurements (e.g. Jasperson 1982) as well as faults and errors in all instruments (e.g. Rust et al. 1990). Belu et al. (2001) explain better correlation between the radar and radiosonde zonal winds than meridional ones due to the latter usually being smaller than the former, and the same absolute errors for the two components results in more significant relative errors for the meridional component.  The authors also compared the winds measured with the CLOVAR windprofiler near London, Canada using DBS technique with winds from the Canadian Meteorological Centre operational model for eight months.
Very good agreement was shown in general, however the radar overestimated the winds relative to the model by 5-20% (more for the meridional than for the zonal component). Comparisons of windprofilers with other models have been carried out.  For example, Gage et al. (1988) found very good correspondence between winds measured with the VHF radar on Christmas Island in the central Pacific and the ECMWF analysis. Schafer et al. (2003) compared winds between 1.5 km and 12 km measured by the windprofilers at four sites in the tropical Pacific between 8 and 13 years to the NCEP-NCAR
reanalysis. Closer agreement was found for the sites where radar data or/and data of nearby located rawinsondes were assimilated by the model.

Our results of inter-comparison of the ESRAD FCA winds and winds from radiosondes reveal systematic underestimation by the radar that is larger for the meridional component (~25%) than for the zonal one (~8%). We also found that ESRAD underestimates the total wind magnitude by ~11%, which is somewhat higher than that found by Reid et al. (2005). Similar
underestimates were found in the comparison between ESRAD and HARMONIE. An analysis of the ability of the full-correlation analysis technique to determine true winds, using synthetic data, has been reported by Holdsworth and Reid, (1995). One part of that study addressed the so called 'triangle effect' whereby winds could be underestimated by an amount which increased with decreasing size of the triangle between the spaced antenna groups used for the analysis. This was found to be due to noise in the detected signals and could be largely corrected by renormalising the cross-correlation functions
between the antenna groups.  Renormalisation is applied in the FCA analysis at both ESRAD and MARA. At ESRAD,

analyses using smaller spacings between antenna groups are also made routinely. These show bigger underestimates of wind speed than the results shown in Fig. 3 and Fig. 5, so the 'triangle effect' is clearly present despite the renormalisation. The renormalisation can be applied correctly only if the noise is random (i.e. all of the noise appears in the zero lag of the autocorrelation functions) and it appears that this is not the case at ESRAD which is in an environment with high levels of

310 RF interference, which also vary over time. Since the baseline BC (32 m) in Fig. 1 is shorter than AB and AC (each 39.4 m), the underestimate in windspeed is most in that direction (BC), which is very close to meridional. The noise levels at MARA are lower and dominated by galactic noise, which is random, so that triangle size effects should be avoidable. Indeed, the comparison of MARA winds with radiosondes in Fig. 8 shows no systematic underestimate of either wind component for MARA winds.

When MARA is compared with the ECMWF-ERA5 reanalysis over a several months period there is some indication that the radar mostly measures slightly smaller winds in the troposphere and lower stratosphere. This might result from limitations in the ability of the model to provide a good description of wind at that particular location which, in turn, might depend on how many local wind data e.g. from radiosondes were assimilated in the model. Neither radar's winds have been assimilated by ECMWF during the comparison periods. Wind information in Antarctica used in the ECMWF model is obtained by

application of an advanced four-dimensional variational data assimilation methodology (Rabier et. al, 1998) in combination with use of radiosondes, satellite-based atmospheric motion vectors and radiances from polar orbiting satellites. The data from radiosondes at only few coastal Antarctic stations are available on regular basis (http://weather.uwyo.edu/upperair/sounding.html). Novolazarevskaya station located 4 km from MARA is just one of them and has not provided radiosoundings since June 2018. Nevertheless, we found surprisingly very good agreement between the

MARA and ECMWF model winds (correlation of 92-95% and bias less than 0.5 m·s$^{-1}$). In the Arctic a lot of different types of observations, including radiosondes, are used within the MetCoOp HARMONIE-AROME modeling system (Muller et al., 2017). Three radiosonde stations: Luleå (69.32°N, 16.13°E), Sodankylä (67.37°N, 26.65°E) and Andøya (69.31°N, 16.13°E) are located within 300 km from Kiruna. Again, ESRAD and HARMONIE winds above 2 km height show good agreement (correlation of 95% and small biases), especially after allowing for the 'triangle size' underestimate by the radar.

In the altitude-resolved comparison between ESRAD and the HARMONIE model as well as radiosondes (Fig. 3 and Fig. 5) we found that below about 2 km the agreement is not good. This is due to technical limitation of ESRAD and other radars, which use the same antenna array for transmission and reception, for measurements at the lowest heights where a received signal from lower heights can be contaminated by low-level 'ringing' after the pulse transmission and by echoes from near-by objects through antenna side-lobes. For MARA we used a small additional receiving-only array that allows accurate

derivation of winds at the lower altitudes too that are in good agreement with the ECMWF model (Figs. 11 and 12). The ESRAD remote receive-only array deployed for the same purpose has had time synchronization problem during the period of interest and these data were not included in our analysis.

## 5 Summary and outlook

The performance of two MST radars: ESRAD in Kiruna, Swedish Arctic and MARA at Maitri, Antarctica in measuring
horizontal winds in the troposphere and lower stratosphere has been evaluated by comparison with radiosondes and NWP
models. The inter-comparison with 28 radiosondes launched from January 2017 to August 2019 showed that the ESRAD
FCA method underestimates zonal and meridional winds by about 8% and 25 %, respectively.  We argue that the ESRAD
receiver group arrangement used for the FCA together with a high level of non-white noise is the likely cause of this
difference. At ESRAD the standard deviation of radar winds in 1-hour averaging bins was 2-2.5 m/s in each component and,
after correcting for the systematic underestimate, the standard deviation of differences between radar and sonde winds was
4.4 m/s (4.8 m/s) for the zonal (meridional) components, respectively. The ESRAD winds were also compared with the
winds computed using the regional NWP model HARMONIE-AROME for the period September 2018 – May 2019. We
found again that ESRAD winds are underestimated by 9% and 24% compared to the model, while showing a very high
correlation between ESRAD and model winds.

The MARA winds were compared with 291 radiosondes launched from February to October 2014 at Novolazorevskaya
station located 4 km from Maitri. We found a good agreement for both zonal and meridional components, with the biases,
defined as the mean difference between the radar and sonde winds, close to 0. The MARA random errors (standard deviation
within 1-hour averaging bins) are estimated to be ~2 m/s in each component. The standard deviation of differences between
radar and sonde winds was 3.7 m/s (2.9 m/s) for the zonal (meridional) components, respectively. The MARA horizontal
wind components have been compared with those from the ECMWF ERA5 reanalysis for the period January - December
2019. In general, the MARA FCA winds are in a good agreement with the model winds. However, the radar zonal winds can
be, on average, a bit lager (2%) as well as smaller (6%) than the model ones, varying by height and season. In turn, the radar
meridional winds are generally 8-11% smaller. On the other hand, we would not expect complete agreement since there are
no close-by radiosondes assimilated by ECMWF during the comparison period.

On the basis of this analysis we conclude that both radars ESRAD and MARA provide measurements of horizontal winds in
the troposphere and lower stratosphere of a good quality with reasonably well-known bias and uncertainty. We plan to use
the radars for validation of winds measured by Doppler lidar on the board the Aeolus satellite in a forthcoming study.

## Data availability

ESRAD data are available from PV on motivated request. MARA data can be obtained on reasonable request from SC.
HARMONIE historical forecasts can be ordered via SMHI's open data service: https://www.smhi.se/en/services/open-data/search-smhi-s-open-data-1.81004.
ERA5 is taken from Copernicus Climate Change Service (C3S) (2017): ERA5: Fifth generation of ECMWF atmospheric
reanalyses of the global climate. Copernicus Climate Change Service Climate Data Store (CDS), date of access 20 December
2019 (https://cds.climate.copernicus.eu/cdsapp#!/home).

## Author contribution

SK, EB and PV developed the codes and conducted the data analysis. SH, MK and HK provided the HARMONIE model outputs. SC and KS provided the MARA data. All co-authors discussed the results. EB prepared the manuscript with contribution from all co-authors.

## Competing interests

The authors declare that they have no conflict of interest.

## Acknowledgements

This work was supported by Swedish National Space Agency (grant numbers 125/18, 279/18). ESRAD operation and maintenance is provided by Esrange Space Center of Swedish Space Corporation. The team members at Maitri station for the 38th Indian scientific expedition to Antarctica (ISEA) are acknowledged for making the year round data possible from MARA. The Antarctic logistics division at NCPOR (India) is also acknowledged for providing necessary supports.

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
