# Peer review of "Validation of wind measurements of two MST radars in northern Sweden and in Antarctica"

_Atmospheric Measurement Techniques, 2020_

## Referee Comment (RC1) · Anonymous Referee #1 · 20 Nov 2020

**General comments**

The article compares the wind measurements of two VHF radar systems with wind measurements from radiosondes and corresponding results from model runs to make statements about the quality of the radar wind measurements. This approach is quite legitimate in case of comparison with the results of another measuring device (e.g. radiosondes). The shown comparison of the wind measurements of ESRAD and radiosondes confirms the underestimation of the wind obtained from FCA analyses but with significant differences between the zonal and meridional components. Since these differences are also well reflected in the comparison of the model results, I wonder if it

could be caused by an instrumental effect. A direct, numerical comparison with results from the literature would help to better classify the results found.

The validation of the radar wind measurements of the mobile radar system in Antarctica was only performed in comparison with corresponding model results. In my opinion, no conclusions about the quality of the radar measurement results can be drawn from such a comparison. However, the fact that the results of the measurements and model calculations agree on average, allows to conclude a reliability of the measurement procedure within the scope of the discussed deviations. Nevertheless, the fact that clear differences between the horizontal components were also found in this comparison raises the question of possible causes, which should be dealt with in more detail in the chapter "Discussion".

**Specific comments**

P2 L52: The description of the antenna array, especially the division of the 288 antennas into 12 subgroups which are connected to separate receivers should be accompanied by a sketch or corresponding reference. The reference given later on page 3 (Kirkwood et al., 2010) describes the ESRAD antenna as 6 groups of 4x6 antennas = 144 antennas, which points to the original system description but is different from the number given above. The knowledge about the arrangement of the groups in the antenna field and their assignment to the receivers (spaced antennas) used for FCA analysis is important for the quality assessment of the method.

Table2: I recommend to add information about the pulse length and pulse shape used in the experiments to the table. This helps to better understand the argumentation used in the dicussion (P12, L257).

Table2: Which points are meant by "number of points = 39"?

P3, L75: As already mentioned, the antenna geometry of the antenna groups, i.e. size and distance to each other, should be listed here.

P5, L119: The "poor performance at lower height" should be explained in more detail.

Table3: See my recommendations for Table 2.

P8, L172: I recommend to indicate the distances between the centres of the three adjacent antenna arrays.

P9, L207: I think that the comparison of a model result with corresponding measurements can lead to the statement that the model can reproduce the measurements well. On the other hand, I do not believe that such a comparison can lead to the conclusion that the measurements are accurate.

P12, L246: The statement made here that the comparison of the ESRAD-FCA wind measurements with those of the radiosondes is consistent with results from the literature should be supported by corresponding concrete examples (figures or numbers and references). The reference to Reid et al. (2005) at the beginning of the discussion lists differences in wind speed in the comparison between radar and radiosonde measurements. Since this is based on more than 3000 measurements, perhaps a comparison of wind speeds (magnitude) should be added to this study.

P12, L257: The specification of pulse length and form in tables 2 and 3 can be used here for a more detailed explanation.

P12, L260: The information about a separate antenna field used for data acquisition is missing in the system description 3.1 and should be added there, possibly accompanied by a sketch.

Summary: I recommend that the differences between the deviations of the zonal and meridional components and possible causes, which in my opinion are significant, should also be addressed here.

---

## Referee Comment (RC2) · Anonymous Referee #2 · 5 Jan 2021

Validation of wind measurements of two MST radars in northern Sweden and in Antarctica

by

Evgenia Belova , Peter Voelger , Sheila Kirkwood, Susanna Hagelin , Magnus Lindskog , Heiner Körnich, Sourav Chatterjee , and Karathazhiyath Satheesan

The manuscript presents wind comparison of the MST radars ESRAD and MARA located at the Arctic and Antarctic in the framework of the Aeolos validation activities. Radar winds are obtained from the full correlation analysis. The authors present several comparison with various validation data sets e.g., radiosondes, reanalysis data

and NWP models. The results are in agreement with previous studies employing all types of MST radars including Doppler beam Swinging methods. The reviewer has some comments, which require some clarification.

Major comment:

The reviewer is a bit concerned about the use of reanalysis data for the comparison. As far as the reviewer is aware, ESRAD is part of the E-Profile network, and, thus the data is available for data assimilation into ECMWF and ERA data products. This needs to be somewhere mentioned in the paper. Although it is likely that only the radiosondes are included in the reanalysis, considering the bias of the radar winds. The reviewer recommends including a statement about a potential data assimilation in NWP.

The reviewer is a bit wondering about the scattering of the MARA winds compared to ERA5. In the paper it is mentioned that there were no radiosondes available for the Antarctic stations around MARA. In fact, what observational data enters ERA5 in the Antarctic, if radiosondes are not available? Can the authors provide a comment on that? Are the MARA winds assimilated?

---

## Author Comment (AC1) · 2 Feb 2021

Title: Validation of wind measurements of two MST radars in northern Sweden and in Antarctica Author(s): Evgenia Belova et al. MS No.: amt-2020-405 MS type: Research article

We thank both referees for their comments that help us to correct and improve our paper. The referee comments are in black, our reply is in blue and changes in the manuscript are in magenta. The new figure numbers are provisional and will be changed when inserting in the final revised manuscript.

**Anonymous Referee #1**

**General comments**

The article compares the wind measurements of two VHF radar systems with wind measurements from radiosondes and corresponding results from model runs to make statements about the quality of the radar wind measurements. This approach is quite legitimate in case of comparison with the results of another measuring device (e.g. radiosondes). The shown comparison of the wind measurements of ESRAD and radiosondes confirms the underestimation of the wind obtained from FCA analyses but with significant differences between the zonal and meridional components. Since these differences are also well reflected in the comparison of the model results, I wonder if it could be caused by an instrumental effect. A direct, numerical comparison with results from the literature would help to better classify the results found.

The validation of the radar wind measurements of the mobile radar system in Antarctica was only performed in comparison with corresponding model results. In my opinion, no conclusions about the quality of the radar measurement results can be drawn from such a comparison. However, the fact that the results of the measurements and model calculations agree on average, allows to conclude a reliability of the measurement procedure within the scope of the discussed deviations. Nevertheless, the fact that clear differences between the horizontal components were also found in this comparison raises the question of possible causes, which should be dealt with in more detail in the chapter "Discussion".

We investigated possible reasons of the significant differences between the zonal and meridional components when comparing the ESRAD and radiosonde winds and came to conclusion that it is due to 'triangle size effect' (Holdsworth and Reid, 1995). We argue that the receiver group arrangement used for the FCA together with a high level of non-white noise is the likely cause of such difference for ESRAD. We will address this issue in the Discussion section of the revised paper.

We agree with the reviewer that comparison of MARA wind with model (ECMWF ERA5) cannot allow making conclusion on accuracy of the radar measurements. Therefore, we will complement the revised manuscript with a comparison of MARA and radiosondes for 2014 where regular radio-soundings from the nearby station Novolazarevskaya were available.

Changes in the manuscript. P9 L186 -> 3.2 MARA versus radiosondes After MARA was deployed at Maitri in 2014, the radar winds were validated using radiosondes launched from the nearby (4 km to the east) Russian station Novolazarevskaya. However, since July 2018 the radio soundings were interrupted and not started again so far. We present here comparison of MARA with radiosondes launched between 08 February 2014 and 30 October 2014 (291 occasions). Radiosonde winds were retrieved from the international database at Univ. Wyoming (http://weather.uwyo.edu/upperair/sounding.html). On average, radiosonde winds were available at 21 heights between the limits (700 - 11000 m) suitable for comparison with MARA. Sondes were usually launched at 0 UT each day, occasionally also at 12 UT and are compared with 1-hour wind averages 00-01 UT (or 12-13 UT) from MARA. Full correlation analysis 'true' winds from each of the three experiments (Table 3) and both main and remote antenna groups are used, with usual acceptance criteria applied, providing on average 38 comparison points per sonde. The results are presented in Fig. GGG. We also plot there the linear fits as in Fig. 2, and the parameters of the fits together with the bias and correlation are provided in the inserts. In contrast to ESRAD, there is no indication that MARA underestimates the winds compared to the sondes (the slopes of the fits for MARA on sonde are slightly less than 1, for sonde on MARA, slightly more than 1). The bias, defined as the mean difference between the radar and radiosonde winds, is close to zero for both zonal and meridional components.

Fig. GGG. Comparison of the MARA and radiosonde (a) zonal and (b) meridional winds. The linear fits are shown as dashed-dotted lines: the radar on sondes in blue and the sondes on the radar in green. The black dashed straight line corresponds to the case when the radar velocity is equal to the sonde velocity.

**P12 L245 -> we add**

Our results of inter-comparison of the ESRAD FCA winds and winds from radiosondes reveal systematic underestimation by the radar that is larger for the meridional component than for the zonal one. We also found that ESRAD underestimates the wind magnitude by 11% that is somewhat higher than that found by Reid et al. (2005). An analysis of the ability of the full-correlation analysis technique to determine true winds, using synthetic data, has been reported by Holdsworth and Reid, (1995). One part of that study addressed the so called 'triangle effect' whereby winds could be underestimated by an amount which increased with

decreasing size of the triangle between the spaced antenna groups used for the analysis. This was found to be due to noise in the detected signals and could be largely corrected by renormalising the cross-correlation functions between the antenna groups. Renormalisation is applied in the FCA analysis at both ESRAD and MARA. At ESRAD, analyses using smaller spacings between antenna groups are also made routinely. These show bigger underestimates of wind speed than the results shown in Figs. 3-5, so the 'triangle effect' is clearly present despite the renormalisation. The renormalisation can be applied correctly only if the noise is random (i.e. all of the noise appears in the zero lag of the autocorrelation functions) and it appears that this is not the case at ESRAD which is in an environment with high levels of RF interference, which also vary over time. Since the baseline BC (32 m) in Fig. XXX (to be added - see below) is shorter than AB and AC (each 39.4 m), the underestimate in windspeed is most in that direction (BC), which is very close to meridional. The noise levels at MARA are lower and dominated by galactic noise, which is random, so that triangle size effects should be avoidable. Indeed, the comparison of MARA winds with radiosondes in Fig. GGG shows no systematic underestimate of either wind component for MARA winds.

**Specific comments**

P2 L52: The description of the antenna array, especially the division of the 288 antennas into 12 subgroups which are connected to separate receivers should be accompanied by a sketch or corresponding reference. The reference given later onpage 3 (Kirkwood et al., 2010) describes the ESRAD antenna as 6 groups of 4x6antennas = 144 antennas, which points to the original system description but is different from the number given above. The knowledge about the arrangement of the groups in the antenna field and their assignment to the receivers (spaced antennas) used for FCA analysis is important for the quality assessment o the method.

**Reply:**

We completely agree with this comment and will add diagrams of the antenna group arrangements used for the FCA. Changes in the manuscript. P3 L65 ->

Figure XXX. Configuration of the ESRAD antenna field. Each blue cross marks the position of a Yagi antenna in the main array (groups 1-12) and in the 'remote' groups (13-15). Each group 1-15 is connected to a separate receiver. Groups 1-12 are also connected to transmitters. The red triangles indicate the baselines for the FCA.

Table2: I recommend to add information about the pulse length and pulse shape used in the experiments to the table. This helps to better understand the argumentation used in the discussion (P12, L257).

Reply:

The pulse length and shape were added to Table 2.

Table2: Which points are meant by "number of points = 39"?

Reply:

We found several mistakes in Table 2 and 3 that were corrected in the revised manuscript.

P3, L75: As already mentioned, the antenna geometry of the antenna groups, i.e. size and distance to each other, should be listed here.

*Reply: We added new figure XXX.*

P5, L119: The "poor performance at lower height" should be explained in more detail.

**Reply:**

This is explained in the Discussion section P12 LL259-261: This is due to technical limitation of ESRAD and other radars, which use the same antenna array for transmission and reception, for measurements at the lowest heights where a received signal can be contaminated with a transmitted pulse.

Table3: See my recommendations for Table 2.

Done in the revised Table.

P8, L172: I recommend to indicate the distances between the centres of the three adjacent antenna arrays.

We added a new figure and description of the MARA antenna array configuration. Changes in the manuscript. P8 L173

The arrangement of antenna array is shown in Figure YYY. There the red triangle 123 indicates the baselines for the full correlation analysis for the main array. The 'remote' groups 4, 5, 6 are used for very low heights where useful data cannot be obtained from transmitting groups.

Figure YYY. Configuration of the MARA antenna field at Maitri station, Antarctica. Each blue cross marks the position of an antenna, single polarisation, dipoles with reflectors in the main array (groups 1-3) and 3-element Yagis in the 'remote' groups (4-6). Each group is connected to a separate receiver. Groups 1-3 are also connected to transmitters.

P9, L207: I think that the comparison of a model result with corresponding measurements can lead to the statement that the model can reproduce the measurements well. On the other hand, I do not believe that such a comparison can lead to the conclusion that the measurements are accurate.

We agree with this comment and added a comparison of MARA winds with radio-soundings at Novolazarevskaya station for 2014 where both radar and radiosonde data were available (new Fig. GGG).

P12, L246: The statement made here that the comparison of the ESRAD-FCA wind measurements with those of the radiosondes is consistent with results from the literature should be supported by corresponding concrete examples (figures or numbers and references). The reference to Reid et al. (2005) at the beginning of the discussion lists differences in wind speed in the comparison between radar and radiosonde measurements. Since this is based on more than 3000 measurements, perhaps a comparison of wind speeds (magnitude) should be added to this study.

**Reply:**

A comparison between wind speeds as measured with ESRAD/MARA and radiosondes was done and the results were discussed.

*Changes in the manuscript. See the reply to General comments.*

P12, L257: The specification of pulse length and form in tables 2 and 3 can be used here for a more detailed explanation.

We are not sure we understand this suggestion. Obviously, it is not possible to receive from heights less than the pulse-length (or a bit longer depending on the shape). But the problems of reception at low altitude are much more than this as the transmitters can 'ring', albeit at a low level, for some time after the nominal end of the transmitted pulse. Also, echoes from closely objects in the antenna sidelines make reception difficult at short ranges. To try to make this clearer we will replace the phrase on line 259/260 '....where a received signal can be contaminated with a transmitted pulse.' with the following:

**Changes in the manuscript.**

**P12 L259->**

....where a received signal can be contaminated by low-level 'ringing' after the pulse transmission and by echoes from near-by objects through antenna side-lobes

P12, L260: The information about a separate antenna field used for data acquisition is missing in the system description 3.1 and should be added there, possibly accompanied by a sketch.

**Changes and new figure YYY were added. See also the reply to General comments.**

Summary: I recommend that the differences between the deviations of the zonal and meridional components and possible causes, which in my opinion are significant, should also be addressed here.

**Reply:**

We adressed this issue in the Discussion section, see the reply to General comments.

**Anonymous Referee #2**

Major comment:

The reviewer is a bit concerned about the use of reanalysis data for the comparison. As far as the reviewer is aware, ESRAD is part of the E-Profile network, and, thus the data is available for data assimilation into ECMWF and ERA data products. This needs to be somewhere mentioned in the paper. Although it is likely that only the radiosondes are included in the reanalysis, considering the bias of the radar winds. The reviewer recommends including a statement about a potential data assimilation in NWP. The reviewer is a bit wondering about the scattering of the MARA winds compared to ERA5. In the paper it is mentioned that there were no radiosondes available for the Antarctic stations around MARA. In fact, what observational data enters ERA5 in the Antarctic, if radiosondes are not available? Can the authors provide a comment on that? Are the MARA winds assimilated?

ESRAD has at times been a part of E-Profile network but not since October 2018. However, the ESRAD wind data have not been assimilated into the regional HARMONIE-AROME NWP system used for comparison in the paper.

Wind information in Antarctic is obtained through the use of application of an advanced fourdimensional variational data assimilation methodology (Rabier et. al, 1998) in combination with use of radiosondes from coastal stations on Antarctica (Novolazarevskaya station located 4 km from MARA is just one of them and has not provided radiosondes since July 2018), satellite based atmospheric motion vectors and satellite-based radiances from polar orbiting satellites passing over Antarctica. MARA data are not assimilated in the ECMWF models.

We will add this information in the revised manuscript.

Changes in the manuscript.

P12 L247->

Both radar winds have not been assimilated into the models. Wind information in Antarctica used in the ECMWF model is obtained by application of an advanced four-dimensional variational data assimilation methodology (Rabier et. al, 1998) in combination with use of radiosondes, satellite based atmospheric motion vectors and satellite-based radiances from polar orbiting satellites. The data from radiosondes at only few coastal Antarctic stations are available on regular basis (http://weather.uwyo.edu/upperair/sounding.html). Novolazarevskaya station located 4 km from MARA is just one of them and has not provided radiosoundings since June 2018.

Rabier, F., Thepaut, J.-N.and Courtier, P., 1998: Extended assimilation and forecast experiments with a four-dimensional variational assimilation system. Q. J. Roy. Meteorol. Soc., 124, 1861-1888.